# Effects of Knee Injury Length on Jump Inside Kick Performances of Wushu Player

**DOI:** 10.3390/medicina57111166

**Published:** 2021-10-27

**Authors:** Jun-Youl Cha, Ha-Sung Lee, Sihwa Park, Yong-Seok Jee

**Affiliations:** 1Division of Sports & Guard, Howon University, Gunsan 54058, Korea; roksfcha@hanmail.net; 2Department of Education (Physical Education Major), Graduate School of Education of Hanseo University, Seosan 31962, Korea; kamui123@hanmail.net; 3Research Institute of Sports and Industry Science, Hanseo University, Seosan 31962, Korea

**Keywords:** injury experience, jump inside kick, static muscle contractility, dynamic muscle contractility, ankle plantarflexor

## Abstract

*Background and Objectives*: When performing the jump inside kick in Wushu, it is important to understand the rotation technique while in mid-air. This is because the score varies according to the mid-air rotation, and when landing after the mid-air rotation, it causes considerable injury to the knee. This study aimed to compare the differences in kinematic and kinetic variables between experienced and less experienced knee injuries in the Wushu players who perform 360°, 540°, and 720° jump inside kicks in self-taolu. *Materials and Methods*: The participants’ mean (SD) age was 26.12 (2.84) years old. All of them had suffered knee injuries and were all recovering and returning to training. The group was classified into a group with less than 20 months of injury experience (LESS IG, *n* = 6) and a group with more than 20 months of injury experience (MORE IG, *n* = 6). For kinematic measurements, jump inside kicks at three rotations were assessed by using high-speed cameras. For kinetic measurements, the contraction time and maximal displacement of tensiomyography were assessed in the vastus lateralis, vastus medialis, rectus femoris, biceps femoris, gastrocnemius lateralis, gastrocnemius medialis, and tibialis anterior. The peak torque, work per repetition, fatigue index, and total work of isokinetic moments were assessed using knee extension/flexion, ankle inversion/eversion, and ankle plantarflexion/dorsiflexion tests. *Results*: Although there was no difference at the low difficulty level (360°), there were significant differences at the higher difficulty levels (540° and 720°) between the LESS IG and the MORE IG. For distance and time, the LESS IG had a shorter jump distance, but a faster rotation time compared to those in the MORE IG. Due to the characteristics of the jump inside kick’s rotation to the left, the static and dynamic muscle contractility properties were mainly found to be higher in the left lower extremity than in the right lower extremity, and higher in the LESS IG than in the MORE IG. In addition, this study observed that the ankle plantarflexor in the LESS IG was significantly higher than that in the MORE IG. *Conclusion*: To become a world-class self-taolu athlete while avoiding knee injuries, it is necessary to develop the static and dynamic myofunctions of the lower extremities required for jumping. Moreover, it is considered desirable to train by focusing on the vertical height and the amount of rotation during jumping.

## 1. Introduction

Wushu is an elite sport that is also practiced as a Chinese martial art. The number of Wushu practitioners in Korea is slowly increasing and many athletes have won medals since Wushu was established as an official sport in the Beijing Asian Games in 1990. In 2003, the rules, content, and judging method were changed and new competitions, known as routine-taolu and self-taolu, were introduced [1,2]. In 2005, the Macao East Asian Games began to apply self-taolu, which allows individual athletes to showcase their own skills and abilities in a creative way. The competition of self-taolu should represent the most effective movements in a short amount of time for each event on a specialized 8 m × 14 m carpet [3]. The scoring method involves ten judges who are divided into Groups A, B, and C. Group A can score five points (quality of operation), Group B can score three points (level of performance), and Group C can score two points (1.4 points for technical difficulty, and 0.6 points for connection difficulty). The level of difficulty, divided into A, B, and C levels, is critical in determining the competition performance of the jump kick motions. In order to obtain a high score in the self-taolu competition, it is necessary to be able to perform techniques of high skill levels. For doing this, many spins must be performed in the air. To date, the spinning skills in self-taolu consist of 360°, 540°, and 720° rotations in the air [4,5].

Jump actions in the self-taolu are skills that include running movements just before jumping, kicking in mid-air, landing posture after performing the jump kick, and quick connection with the following technique immediately after landing. Most injuries occur during landing after jumping, and the knee is reported to be the most common among lower extremity joints [6]. Meanwhile, mid-air moves can only be performed well if an athlete has learned scientific training methods and acquired skills according to other difficult moves. For the kinematic aspects of mid-air motions, the velocity change of the body center in performing the 540° rotation was analyzed. Aerial motion can only show altitude and the internal rotation of the left and right lower extremity joints around the body vertical axis at the highest apex [3,7]. It is desirable to effectively utilize ground reactions during rolling, and the right foot should not distribute forces in the left and right directions to lower the center of gravity and increase the height of the jump [5].

The duration of being in mid-air, the rotation of the upper body before jumping, and the action of the outside or inside kicks during the jump are important in determining the level of difficulty of the jump inside kick [8]. A study reported that it is beneficial to increase acceleration by applying the ground reaction force in the hop motion for the jump outside kick [7]. Increasing the force on the footplate by lengthening the hop time also makes it possible to efficiently ascend into the air with greater force [9,10]. It is important to understand the techniques, such as running, hopping, ground reaction force, jumping, and spinning time, as well as the kinematic posture of high-ranked athletes to achieve the required physiological abilities. However, there is a lack of evidence in the literature regarding the analysis of the Wushu jump inside kick for self-taolu. Moreover, there is a lack of research on exercise performance in Wushu players, especially the myofunction of the lower extremities, of athletes who returned to the field after a knee injury.

Therefore, this study investigated the performance of 360°, 540°, and 720° jump inside kicks in Wushu self-taolu for athletes who had injuries related to the aerial rotation technique. In addition, comparative physiological analyses of the static and dynamic myofunctions of the athletes were undertaken to help young athletes perform high-latitude aerial rotation techniques while avoiding damage. The major research hypotheses addressed in this thesis are as follows: First, there would be differences in the 360°, 540°, and 720° rotations of jump inside kicks depending on the injury length of knee joints. Second, there would be differences in the static and dynamic contractile myofunctions depending on the injury length.

## 2. Materials and Methods

### 2.1. Participants

The participants of this study were selected from male Wushu national athletes who could perform 360°, 540°, and 720° rotation drills. They were all right-handed and right-footed. Their mean (SD) age was 26.12 (2.84) years old. All of them had suffered knee injuries and were all recovering and returning to training. All players had no cardiovascular problems and were excluded if they had taken any treatment or medication known to affect physical condition or had undergone any major surgery except for a knee joint operation during the one year before the start of this study. The following were also reasons for exclusion: having a history of cerebrovascular disease, impairment of a primary organ system, severe lung disease, cerebral trauma, uncontrolled hypertension, or psychiatric disorder. Twelve players were enrolled in this study. After taking baseline measurements, the group was classified into a group with less than 20 months of injury experience (LESS IG, *n* = 6) and a group with more than 20 months of injury experience (MORE IG, *n* = 6). The reason for classifying the group as 20 months was the application of the Cochran–Mantel–Haenszel equation, which expresses the injury risk as the number of knees injured as a percentage of the total number [11]. The subjects of this study were evaluated by a specialist, and athletes with an injury length of 20 months or more were classified as having a high risk of repeated injury, whereas those of less than 20 months were classified as having a moderate injury length. In LESS IG, three patients were diagnosed with right anterior criuciate ligament (ACL) rupture, but recovered after 5, 7, and 8 months of rehabilitation treatment, respectively. On the other hand, one player was diagnosed with a right hamstring sprain and received 5 months of treatment, while the other two players recovered after 18 and 20 months of postoperative rehabilitation treatment for partial rupture of the right knee joint cartilage, respectively. In the MORE IG, two players were diagnosed with a simple right ACL rupture and two players were diagnosed with ACL+PCL (posterior criuciate ligament) complex rupture of the right knee joint, and recovered after 7, 14, 16, and 18 months of rehabilitation treatment, respectively. Meanwhile, one player was diagnosed with a right hamstring rupture and recovered after 12 months of treatment. However, 7 days later, during training, he ruptured again in the same area and had to undergo treatment for another 12 months. The other player was diagnosed with an ACL+PCL complex rupture of the right knee and recovered after 18 months including surgery and rehabilitation. However, this athlete recovered after 18 months of rehabilitation after surgery due to a partial rupture of the right knee joint cartilage during training. Table 1 shows the complete characteristics of the participants.

### 2.2. Experimental Design

This study followed the principles of the Declaration of Helsinki and received approval from the institutional ethics committee. Prior to the study, the investigator explained all the procedures to the players who were recruited through advertisements and written informed consent was obtained before enrollment. All players arrived at the research center to sign an informed consent form, complete a self-report questionnaire about their health status included in the physical examination, and to take assessments that consisted of body composition, jump motion, tensiomyography (TMG), and isokinetic moments tests.

### 2.3. Measurement Methods

#### 2.3.1. Body Composition Measure

An Inbody 230 (Biospace Co., Ltd., Seoul, Korea) analyzer with the bioelectrical impedance analysis method was used for the body composition measurements of all Wushu players. This analyzer is a segmental impedance device that assesses the voltage drop in the upper and lower body. The participants were asked to remove all metal objects and anything else that might interfere with the electric stimuli, including socks, before stepping onto the platform. They were also asked to hold onto the handles and stand still for around 3 min [4,12]. Bioelectrical impedance analysis can track fat mass by conducting high frequency (500~800 KHz) harmless to the human body and using the difference in electrical resistance between adipose tissue and non-adipose tissue. In order to minimize the error, in this study, food intake was abstained 4 h before the test, and alcohol was prohibited 48 h before the test. In addition, exercise was not allowed 12 h before the test. On the other hand, it was necessary to urinate 30 min before the test.

#### 2.3.2. Kinematic Motion Measure

Two-dimensional imaging was performed to analyze the movements of a Wushu jump inside kick at 360°, 540°, and 720°, respectively. Wushu players all took the jump inside kick in a counterclockwise direction. To obtain spatial coordinates for image analysis, a control point frame with three control points was installed with a width of 1 m, a height of 2 m, and a length of 2 m; therefore, all the players’ movements were captured, as shown in Figure 1.

The point frame was photographed for about 1 s, after which the control point frame was removed. Image analysis was performed using two high-speed cameras (FDR-AX100, Sony, Japan), and analyzed with Dartfish (Dartfish Live S10 program, Dartfish, Switzerland). A total of two units were installed to cover all the players’ performance. For the experiment, each player engaged in warm-up and kicking practice and had three markers attached to each joint area. Then, they performed five drills while being photographed, and the motions that were determined to be the most complete were selected and analyzed.

As shown in Figure 2, five events and four phases were used for the kinematic analysis of three motions of the jump inside kick. Event (E) 1 indicates the preparation stage. E2 indicates the moment of the knee joint of the lower extremity reaching the minimum angle. E3 indicates the moment that the lower leg leaves the ground. E4 indicates the moment the kicking leg reaches the highest point and makes impact with a hand. E5 indicates the moment the foot lands on the ground. Phase (P) 1 indicates the preparation segment from E1 to E2. P2 indicates the take-off segment from E2 to E3. P3 indicates the kicking action from E3 to E4. P4 indicates the turning/spinning and landing segments from E4 to E5. After recording the entire sequence of events using a camera, the anatomical coordinate points of each frame were digitized.

#### 2.3.3. Kinetic Variables Measures

##### TMG Measure for Static Muscle Contraction

This study employed a TMG device (TMG100, TMG-BMC Ltd., Ljubljana, Slovenia). The extraction of contractile parameters from TMG responses is straightforward and does not require special post-processing of filtering [13,14]. The Wushu players were lying in a prone position on the examination couch. The correct angles in the joints allowing for relaxation of the examined muscles were ensured using guidelines and recommendations of the device manufacturers (GK 40 Panoptik d.o.o., Ljubljana, Slovenia). Two adhesive electrodes stimulating the muscle were placed 2–5 cm apart. The electrodes were placed in a way that did not affect the tendons and allowed the contraction of the particular muscle to be isolated and the simultaneous activation of nearby muscles to be avoided. In the TMG test, the uninjured leg was examined first, and then the leg in the injured area was examined. The placement of the sensor was selected in order to locate the thickest part of the muscle. The sensor was applied to the skin halfway between the electrodes [12]. The electrodes received one 1-millisecond single-phase rectangular pulse from the electrostimulator inducing percutaneous muscle contraction. The pulse power was gradually increased by 10 mA until the maximal contraction reaction was achieved. For minimizing the effects of fatigue, 10 s intervals were taken between the pulses. Typical maximum contraction reactions were recorded between 40 and 80 mA [15]. Displacement–time curve recordings allow muscle contractile properties to be assessed for contraction time (Tc) and maximal displacement (Dm) [13,16,17]. This study measured the Tc and Dm of TMG in the vastus lateralis, vastus medialis, rectus femoris, biceps femoris, gastrocnemius lateralis, gastrocnemius medialis, and tibialis anterior in both legs, as shown in Figure 3.

##### Isokinetic Measure for Dynamic Muscle Contraction

All players were positioned in an isokinetic dynamometer (HUMAC^®^/NORM^TM^ Testing and Rehabilitation System, CSMi, MA, USA) according to the manufacturer’s guidelines, as shown in Figure 4. Testing was performed on the uninjured side first, and then performed on the non-injured side.

All players for the knee extension/flexion test were submitted to a warm-up program before the test. Each participant was placed in the equipment’s adjustable seat. The tested limb was placed and fixed with a Velcro strap on a support over the quadriceps, and the knee joint was positioned at 90°. After the participant was positioned and uniformly stabilized, the participant’s leg was statically weighed to provide for gravity compensation. Each participant was concentrically tested at 60°/s and at 180°/s. The range of motion (RoM) of extension/flexion was at 0° and 90°. Players then performed 4 maximal warm-up repetitions and 5 maximal test repetitions for evaluating peak torque (Pt) and work per repetition (Wr) at 60°/s, and performed 15 maximal test repetitions for evaluating Pt, fatigue index (Fi), and total work (Tw) at 180°/s. The rest time between the two angular speeds was 60 s.

The players also took the ankle inversion/eversion tests [17]. Each participant was placed in an adjustable seat. The tested limb was placed and fixed with a strap on a support under the gastrocnemius, and the knee was positioned at 30° of flexion. The foot was placed on the inversion/eversion apparatus and fixed with two straps. The axes of the ankle were positioned according to the placement proposed by research [18]. The trunk was stabilized with constraining straps, and an extra strap was used to stabilize the hip at 80° flexion. The arms and the lower limb that was not being tested were placed in a resting position. Two RoM targets consisting of plastic markers were placed at the level of the footplate to facilitate inversion/eversion movements [18]. The players performed 4 submaximal practices at each test speed (60°/s and 90°/s); each test comprised of movement from ankle eversion (40°) to inversion (55°). The measured results of isokinetic moments at 60°/s were analyzed by Pt and Wr. The measures at 90°/s were analyzed by Pt, Fi, and Tw. A rest period of 60 s was given between the tests.

For the ankle plantarflexion/dorsiflexion test, the players were placed in the prone position with the knee placed in full extension and stabilized at the level of the distal thigh with straps. The testing side was placed on the footplate attachment and the lever arm of the dynamometer was aligned with the foot. The motor axis was positioned against the lateral malleolus. The RoM test for both plantar- and dorsiflexions was set during the first test. The players performed 4 submaximal practices at each test speed (30°/s and 120°/s); each cycle comprised of movements from ankle plantarflexion (50°) to dorsiflexion (20°). In the concentric test mode, the players were instructed to push the lever arm along the full angular sector in both directions [19,20]. Both ankles of each participant were assessed and the same rest time was given. The measured isokinetic moments at 30°/s were analyzed by Pt and Wr. The measures at 120°/s were analyzed by Pt, Fi, and Tw.

### 2.4. Data Process and Statistical Analyses

The sample size was calculated using G Power Software version 3.1.9.7 [21]. The necessary sample size of 12 subjects was calculated from data to achieve a power of 0.40 and an effect size of 0.90, with an α level of 0.05. All data were reported as mean ± standard deviation and carried out using the IBM^®^ SPSS^®^ Statistics software (version 22.0. IBM Corporation; Armonk, NY, USA). The distribution of all data was checked using the Shapiro–Wilk test. Due to non-normally distributed data, a non-parametric Mann–Whitney U test was conducted. Effect sizes were determined by converting partial eta-squared (ES) to Cohen’s *d* [22]. For all analyses, the significance level was set at *p* ≤ 0.05.

## 3. Results

### 3.1. Analysis of Demographic Variables

As shown in Table 1, although the injury period was a significant difference between the groups, there were no significant differences in the remaining variables of the two groups. The demographic variables of this study indicated the homogeneity of the subjects.

### 3.2. Kinematic Analysis of Jump Inside Kick

#### 3.2.1. Comparisons of Event Time and Phase Time of Jump Inside Kick

Table 2 shows that the total event time from E1 to E5 at 360° in the LESS IG appeared to be 0.04 s longer than that of the MORE IG. On the other hand, at 540°, the total event time from E1 to E5 in the LESS IG was 0.27 s shorter than that of the MORE IG, indicating that the ability of the LESS IG to rotate in the air was faster than that of the MORE IG. These results were similar at 720°. The results show that the event times of rotation in the LESS IG was significantly faster than those of the MORE IG for 540° and 720° jump inside kicks.

Table 2 also shows that the total phase times at 360° and at 540° did not show a significant difference between the two groups, but at 720° rotation, it was confirmed that the LESS IG was significantly faster than the MORE IG.

#### 3.2.2. Comparisons of Distance and Time of Jump Inside Kick

Table 3 represents the distance and time from take-off to landing between the LESS IG and the MORE IG. For the total distance covered in the 360° jump inside kick, the LESS IG was shorter than the MORE IG. This trend was similar in 540° and 720° rotation, and it can be observed that there is a statistically significant difference between the groups.

### 3.3. Kinetic Analysis of Jump Inside Kick

#### 3.3.1. Analysis of Static Muscle Contractions in Lower Legs

Table 4 shows that there was a significant difference in Tc between the LESS IG and the MORE IG. The Tcs of the right gastrocnemius lateralis, right gastrocnemius medialis, and right tibialis anterior of the LESS IG were significantly higher than those of the MORE IG. Meanwhile, the Tcs of left rectus femoris, left tibialis anterior and left vastus medialis of the LESS IG were significantly higher than those of the MORE IG. Dm also showed a similar tendency to Tc. This tendency can be seen as a phenomenon that occurs because the Wushu athletes must rotate to the left with the center point on their left leg in order to perform a jump inside kick.

#### 3.3.2. Analysis of Dynamic Muscle Contractions in Isokinetic Low Angular Speed

Table 5 shows that there were significant differences in the isokinetic knee extensor and flexor at 60°/s between the LESS IG and the MORE IG. Specifically, Pt and Wr in the left extensor and flexor of the LESS IG were significantly higher than those of the MORE IG. A similar tendency was observed in the ankle invertor/evertor, although there were no significant differences between both groups. In the ankle plantarflexion/dorsiflexion test at 30°/s, the Pt and Wr in the right plantarflexor of the LESS IG were significantly higher than those of the MORE IG.

#### 3.3.3. Analysis of Dynamic Muscle Contractions in Isokinetic Moderate to High Angular Speed

Table 6 shows that there were significant differences in isokinetic knee extensor and flexor at 180°/s between the LESS IG and the MORE IG. The knee extensor that serves as a crutch when rotating to the left was significantly higher in the left leg of the LESS IG, whereas the knee flexor that acts as a crutch when performing a jump was significantly higher in the right leg. A similar tendency was also observed in the isokinetic ankle invertor/evertor and ankle plantarflexor/dorsiflexor.

## 4. Discussion

The findings of this study revealed that when comparing the Wushu self-taolu athletes who have had different injury periods, there was no significant difference in body composition and the 360° jump inside kick. However, there were significant differences between the groups for the 540° and 720° jump inside kicks. In addition, in the event and phase time related to rotational time, there were no differences between the groups for the 360° jump inside kick, but there were significant differences between the groups at the higher difficulty levels. In other words, the LESS IG had a shorter jump distance, but faster rotational time, although there was no significant difference in body composition compared to the MORE IG. Specifically, this study found that there were significant differences between the groups in the static and dynamic myofunctions between the LESS IG and the MORE IG.

When the jump inside kick was performed, the time in mid-air for the LESS IG was longer than that of the MORE IG prior to completing the inside kick. In this study, it is thought that the LESS IG efficiently performed the jump with sufficient space and elapsed time during the inside kick motions at 360°, 540°, and 720°. The distance and time from take-off to landing between the LESS IG and the MORE IG also showed that the LESS IG had a shorter jump distance, but a faster rotational time; such a fast rotation time must be supported by the myofunction capable of exerting force. The jump inside kick is a technique that starts with running from a distance, jumping counterclockwise while jumping (Phases two to three), and kicking with the inside surface of the right foot while jumping (Event four). In other words, touching the sole of the right foot during the three types of jump inside kicks occurred close to the end of Event four. For a better jump front kick, an athlete needs to be able to jump vertically. For this purpose, the forward speed just before hopping should be fast; during the hopping period, it is important to shift this forward speed vertically [6]. The better the athlete is, the shorter the hop time in the run before and after the jump is; the shorter the intersection between the vertical and horizontal speeds in the section immediately before and after the jump is, the more efficient the performance is [23].

At international competitions, there are many foreign Wushu athletes who perform at C level, and the results are excellent. In Korea, there are only few athletes who can perform at C level. Self-taolu athletes must perform advanced skills in order to excel in the Asian Games and World Championships. Therefore, this study intended to present the analysis and application method for performing high-difficulty techniques. Recently, many quantified data and instruction models have been proposed through these analyses. Most studies have only analyzed difficulty levels for A and B, such as the 360° and 540° jump inside kicks, while research on C-level techniques, such as the 720° jump inside kick, is rare. When performing the three types of jump inside kicks, the LESS IG showed superior time and distance compared to the MORE IG. In this regard, Leporace et al. reported that it is desirable to increase the vertical velocity in the jump motion, and to perform the motion using the rotation of the lower joint on the long axis [6]. This study found that although there were no differences at 360° between both groups, there were significant differences in the technical skills in the 540° and 720° jump inside kicks between the groups. That is, for the higher-ranking levels of Wushu athletes in Korea, more technical training is needed at over 360° spinning skill such as a shorter spinning time [1,4,24].

In order to perform the higher levels of difficulty for the jump inside kick, the muscle contractility of the lower extremities is very important [6,25]. In this regard, this study investigated the static muscle contractility of the lower extremity in the Wushu self-taolu players. In the results of this study, Dm in the right biceps femoris of the LESS IG was significantly lower than that of the MORE IG. This indicates a high muscle tone expressed in the hamstrings of the LESS IG may be a concern in the future due to excessive stiffness. It is also evidence that the training related to jumping is repeated periodically. In addition, the high Tc and low Dm of the right gastrocnemius lateralis of the LESS IG showed a significant difference with the values of the MORE IG, and gastrocnemius medialis of the LESS IG showed similar results. On the other hand, compared to the MORE IG, the low Dm of the left gastrocnemius lateralis of the LESS IG, the high Tc and low Dm of the left rectus femoris, and the high Tc of the left tibialis anterior and vastus medialis can be interpreted to mean that the LESS IG is better trained or rehabilitated for jumping and rotating compared to the MORE IG. In this regard, García-Manso et al. reported that the post-activation potentiation after strength exercises was responsible for the observed changes in muscle response at the end of the first set [26]. This mechanism becomes significant with changes in increased Tc and decreased Dm. Several researchers also suggested that a low Dm indicates a high muscle tone and excessive stiffness in the muscle structures, whereas an elevated Dm indicates a lack of muscle tone or the appearance of muscle fatigue [12,13,26]. Considering that Dm varies depending on the load of strength training, the recovery time between repetitions, and the type of muscle contraction, it can be estimated that the LESS IG analyzed in this study had a higher training volume than the MORE IG. The normal curve from TMG has a steep shape, and Tc appears at short intervals [27]. Characteristically, the shape of the overall curve appears to collapse after the injury, which is due to the fact that muscle contraction does not proceed normally and rapidly [26,27]. The two parameters enable the muscular composition of the studied muscular groups to be increased [28], correlated with the increase in the contraction time, and the decrease in the muscular displacement amplitude [12,29]. These parameters have normal average values for Tc, namely, 32.83 ms, and for all muscular groups, the average Dm value is 8.17 mm [29,30]. In the static muscle contractions in this study, there was no significant difference in Tc between the LESS IG and the MORE IG, but the Tc of the left rectus femoris, the Dm of the right biceps femoris, and the Dm of the gastrocnemius lateralis in the LESS IG were significantly higher than those in the MORE IG. The Tc of the right biceps femoris in the LESS IG was also significantly higher than the MORE IG. In other words, the left rectus femoris, right biceps femoris, and right gastrocnemius lateralis are all muscle groups that are required for leaping and kicking, which are significantly higher in the LESS IG than the MORE IG. In particular, the Tc of right biceps femoris was very high in the LESS IG, and it was found to satisfy the hypothesis of this study. It was also found that Dm was most effective in detecting muscle hypertrophy, and that muscle stiffness secondary to muscle hypertrophy could induce a decrease in the Dm of Wushu self-taolu athletes. These results show the fact that when the injury period is short when landing after an inside jump, it is possible to show a little better performance. In other words, as a Wushu athlete, in order to become a superior athlete, it is necessary to have some level of static muscle function in the lower extremities, and this result is considered to be the only way to prevent injuries [31].

Some of the contractile parameters measured by TMG have been found to correlate with muscle peak torque [32], and with the spatial distribution of fiber types in human muscles [14,16,28]. In addition to this relation, the dynamic phenomenon in sports shows that the role of dynamic myofunction as well as static myofunction is very important [33]. From this point of view, this study also investigated the static as well as the dynamic myofunctions, and the following results were observed. The knee extensor muscles that include the vastus medialis, vastus lateralis, and rectus femoris, fixed the knee joint upon landing after the jump. These muscle groups did not differ significantly between the groups in this study. However, the biceps femoris, an important knee flexor muscle at the time of the jump, appeared very predominantly in the LESS IG. In examining the dynamic muscle contractility, the moments of isokinetic low angular speeds in the LESS IG were higher than those of the MORE IG. Specifically, the Pt and Wr of the left knee extensor and flexor of the LESS IG were significantly higher than those of the MORE IG. Through this myofunction of the thigh, explosive jumps are made, and it helps to maintain balance even when landing after a mid-air rotation. This phenomenon is thought to be the development of extensor and flexor muscles in the center of the knee joint, which must withstand the force applied to the left lower extremity just before the jump when athletes need to jump in the left direction. Moreover, the right ankle plantarflexor, which triggers the jumping force to the left, was significantly higher in the LESS IG than in the MORE IG, which is interpreted to suggest the importance of the right plantarflexor that triggers the jumping time to the left. These results seem to support the results shown in the TMG test presented in Table 4. That is, the Tc of the right gastrocnemius lateralis was 32.69 ± 3.76 ms in the LESS IG, while it was 16.23 ± 24.13 ms in the MORE IG, which is consistent with the result of the dynamic contractility of the right ankle plantarflexor.

As shown in Table 6, the moments of isokinetic moderate to high angular speeds are shown similarly to the moments of low angular speed, although it shows specific results. That is, the Pt of the left knee extensor of the LESS IG was significantly higher than that of the MORE IG, and the Tw also showed a similar result, suggesting that the role of the left lower extremity is important before jumping and during landing [4,6,33]. The Fi in the LESS IG is significantly higher than in the MORE IG, which means that muscle fatigue in the LESS IG is high in the left thigh, confirming the meaning of the results once again [31]. On the other hand, unlike the extensor of the knee, Pt and Tw of the knee flexor of the LESS IG were significantly higher than those of the MORE IG on both the left and right sides, whereas there was no difference in Fi. This result shows that the knee flexor should be excellent in bending the knee joint before jumping and absorbing the shock during landing, while the muscle fatigue should be low. Meanwhile, the Tw of the right ankle plantarflexor in the LESS IG was 594.00 ± 48.08 Nm, whereas that in the MORE IG was 322.50 ± 137.89 Nm, which shows a significant difference between both groups. The Tw of the left ankle plantarflexor in the LESS IG was 520.50 ± 159.10 Nm, whereas that of the MORE IG was 372.50 ± 126.57 Nm, which shows a significant difference between both groups. This result shows that when performing the Wushu jump inside kick, the ankle plantarflexor for extending the ankle should be able to perform myofunction without fatigue for a long time. This study confirmed that the excellent rotational skills of jump inside kicks in the Wushu self-taolu athletes come from the well-equipped static and dynamic muscle contractile properties while minimizing the injury frequency or length. Ultimately, this study found that when Wushu players rotate 360, 540, and 720 degrees in the air, the rotation speed varies according to the three target rotation angles, and there is a significant difference according to the length of the injury. However, when the length of the injuries of the athletes is long, the muscle function of the lower extremities to cause rotation in the air is low, and moreover, when the number of rotations is high, it is more evident. These results suggest that even if an athlete is injured, treatment should be carried out quickly so that the frequency or length of the injury does not occur as often.

## 5. Conclusions

First, the event time, phase time, and distance/time for 540° and 720° jump inside kicks of the LESS IG were higher than those of the MORE IG. Second, there were significant differences in the Tc and Dm between the LESS IG and the MORE IG. The left lower extremity of the LESS IG was well developed. The right biceps femoris was significantly higher in the LESS IG than in the MORE IG. This tendency was similarly observed at isokinetic low to high angular speeds. It can be interpreted that the static and dynamic muscle contractile properties required for the jump inside kicks were significantly higher in the less experienced injury of lower legs in the Wushu athletes. This study confirmed that in order to be a non-injured self-taolu athlete, it is necessary to develop myofunctions of the knee extensor, knee flexor, and ankle plantarflexor for jump inside kicks. In other words, in preparation for injury to the lower extremities that may occur during jumping and landing, muscle function training should be continued. Additionally, it can be concluded that it is desirable to train and to rehabilitate accordingly because there is less torsion of the lower extremities upon landing after a high air jump. However, this study had a small sample size due to the limited number of Wushu players in Korea. In addition, healthy Wushu players were not included in the study. Therefore, there are limitations in being able to generalize these results to other populations and types of competitions.

## Figures and Tables

**Figure 1 medicina-57-01166-f001:**
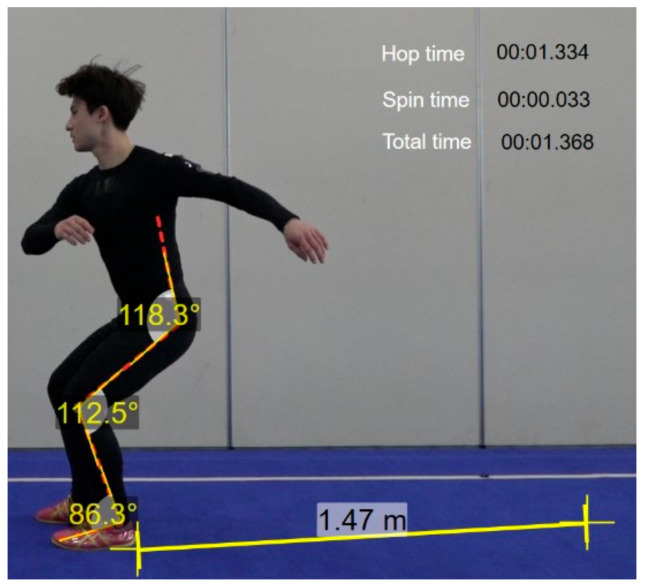
Marker attachment locations. To obtain spatial coordinates for image analysis, a control point frame with three control points was installed. Then, an observer captured images for assessing hop time, spin time, and total time.

**Figure 2 medicina-57-01166-f002:**
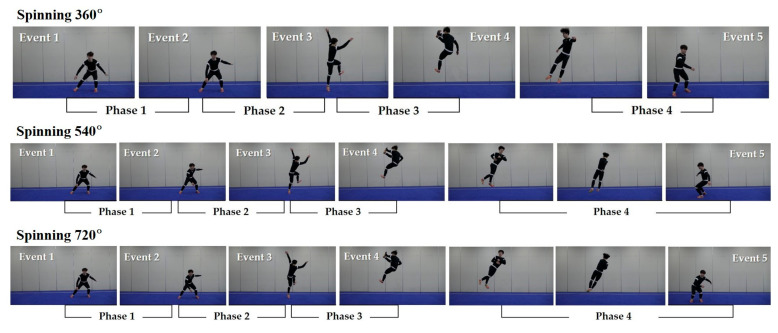
Events and phases in 360°, 540°, and 720° jump inside kicks. Two-dimensional images were performed to analyze the movement of jump inside at 360° (**up**), 540° (**middle**), and 720° (**bottom**). Event (E) 1, preparation stage; E2, moment of knee joint reaching the minimum angle; E3, moment of the lower leg leaving the ground; E4, moment the kicking leg reaching the highest point and making impact with a hand; E5, moment of the foot landing on the ground. Phase (P) 1, preparation segment from E1 to E2; P2, take-off segment from E2 to E3; P3, kicking action from E3 to E4; P4, turning/spinning and landing segments from E4 to E5.

**Figure 3 medicina-57-01166-f003:**
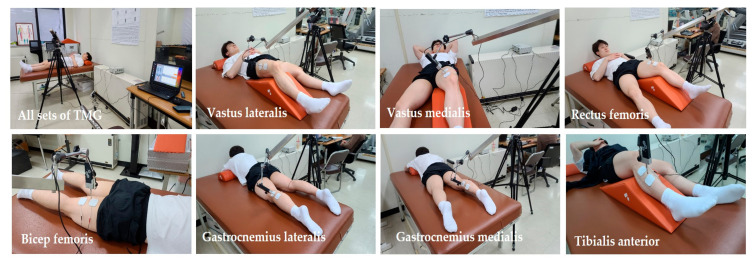
TMG measures’ samples. TMG is a technique based on the quantification of radial muscle belly displacement in response to a single electrical stimulus. The measurements are performed in a relaxed position. A digital transducer measured the radial displacement pressed perpendicularly against the skin above the muscle belly in vastus lateralis, vastus medialis, rectus femoris, biceps femoris, gastrocnemius lateralis, gastrocnemius medialis, and tibialis anterior in both legs.

**Figure 4 medicina-57-01166-f004:**
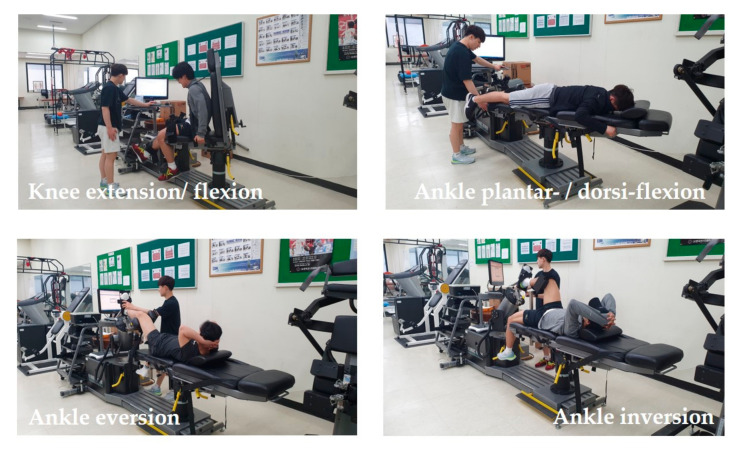
Isokinetic moments measures’ samples. The knee joint for knee extension and flexion was positioned at 90°. All participants were concentrically tested at 60°/s and 180°/s. The test angles for ankle plantarflexion and dorsiflexion comprised of movement from 50° (plantarflexion) to 20° (dorsiflexion), where all participants were concentrically tested at 30°/s and 120°/s. The test angles for ankle inversion/eversion comprised of movement from 55° (inversion) to 40° (eversion, where all participants were concentrically tested at 60°/s and 90°/s.

**Table 1 medicina-57-01166-t001:** Physical characteristics of the Wushu self-taolu players.

	Groups		Z	ES	*p* *
	LESS IG	MORE IG			
Age (y)	25.50 ± 2.87	26.75 ± 2.86	−2.683	0.436	0.115
Height (cm)	169.24 ± 3.64	167.80 ± 5.41	1.455	0.312	0.283
Weight (kg)	65.45 ± 6.76	66.64 ± 5.43	−1.327	0.194	0.395
Athletic career (month)	192.00 ± 51.47	200.00 ± 37.69	−0.167	0.177	0.937
Injury period (month)	8.17 ± 4.69	20.00 ± 8.85	−2.330	1.670	0.015

All data represent mean ± standard deviation. * Analyzed using Mann–Whitney U test. LESS IG, a group with less than 20 months of injury experience; MORE IG, a group with more than 20 months of injury experience. ES, effect size; Z, statistical symbol.

**Table 2 medicina-57-01166-t002:** Event time and phase time of jump inside kick.

		Groups			
		LESS IG	MORE IG	Z	ES
Event 360°	Event 1 (s)	0.00 ± 0.00	0.00 ± 0.00	-	-
	Event 2 (s)	0.20 ± 0.04	0.20 ± 0.01	-	0.00
	Event 3 (s)	0.42 ± 0.02	0.40 ± 0.10	−1.000	0.277
	Event 4 (s)	0.70 ± 0.00	0.68 ± 0.02	−1.000	0.277
	Event 5 (s)	1.07 ± 0.00	1.07 ± 0.00	*-*	*-*
	Total (s)	2.39 ± 0.07	2.35 ± 0.02	−0.775	0.777
Event 540°	Event 1 (s)	0.00 ± 0.00	0.00 ± 0.00	*-*	*-*
	Event 2 (s)	0.18 ± 0.03	0.23 ± 0.00	−1.633	0.777
	Event 3 (s)	0.38 ± 0.02	0.45 ± 0.02	−1.549	3.500
	Event 4 (s)	0.62 ± 0.02	0.72 ± 0.02	−2.549 *	5.000
	Event 5 (s)	1.05 ± 0.03	1.10 ± 0.00	−1.633	5.000
	Total (s)	2.23 ± 0.01	2.50 ± 0.05	−2.562 *	7.488
Event 720°	Event 1 (s)	0.00 ± 0.00	0.00 ± 0.00	*-*	*-*
	Event 2 (s)	0.17 ± 0.05	0.17 ± 0.00	*-*	7.488
	Event 3 (s)	0.36 ± 0.00	0.40 ± 0.00	−1.633	7.488
	Event 4 (s)	0.62 ± 0.02	0.63 ± 0.00	*-*	7.488
	Event 5 (s)	1.07 ± 0.05	1.05 ± 0.02	0.001	0.525
	Total (s)	2.11 ± 0.11	2.75 ± 0.02	−2.532 *	8.095
Phase 360°	Phase 1 (s)	0.12 ± 0.16	0.20 ± 0.00	0.001	8.095
	Phase 2 (s)	0.19 ± 0.02	0.20 ± 0.00	-	8.095
	Phase 3 (s)	0.34 ± 0.09	0.32 ± 0.02	0.001	0.306
	Phase 4 (s)	0.54 ± 0.23	0.39 ± 0.02	−0.408	0.918
	Total (s)	1.17 ± 0.14	1.10 ± 0.04	−1.608	0.679
Phase 540°	Phase 1 (s)	0.10 ± 0.14	0.23 ± 0.00	−2.633 *	0.679
	Phase 2 (s)	0.20 ± 0.05	0.22 ± 0.03	−0.775	0.485
	Phase 3 (s)	0.32 ± 0.12	0.27 ± 0.01	0.001	0.587
	Phase 4 (s)	0.55 ± 0.11	0.39 ± 0.02	−2.549 *	2.023
	Total (s)	1.16 ± 0.04	1.10 ± 0.00	−1.633	2.023
Phase 720°	Phase 1 (s)	0.10 ± 0.14	0.17 ± 0.00	0.001	27.457
	Phase 2 (s)	0.15 ± 0.02	0.23 ± 0.00	−2.633 *	27.457
	Phase 3 (s)	0.32 ± 0.06	0.24 ± 0.00	−2.651	27.457
	Phase 4 (s)	0.14 ± 0.09	0.37 ± 0.05	−2.549 *	3.159
	Total (s)	1.10 ± 0.01	2.09 ± 0.05	−2.862 *	27.457

All data represent mean ± standard deviation. LESS IG, a group with less than 20 months of injury experience; MORE IG, a group with more than 20 months of injury experience. ES, effect size; *, *p* < 0.05.

**Table 3 medicina-57-01166-t003:** Distance and time required by events of jump inside kick.

			Groups			
			LESS IG	MORE IG	Z	ES
360°	DPB	d (m)	2.19 ± 0.99	2.94 ± 0.42	−0.775	0.986
		t (s)	1.72 ± 0.55	2.59 ± 0.73	−0.786	1.346
	DSL	d (m)	0.95 ± 0.74	1.66 ± 0.68	−0.775	0.999
		t (s)	0.98 ± 0.21	0.77 ± 0.14	−0.775	1.176
	TDPL	d (m)	3.14 ± 1.73	4.60 ± 1.10	−2.973 *	1.007
		t (s)	2.70 ± 0.33	3.35 ± 0.59	−2.775 *	1.359
540°	DPB	d (m)	2.51 ± 1.00	2.89 ± 0.11	0.001	0.534
		t (s)	1.49 ± 0.45	1.82 ± 0.12	−0.775	1.002
	DSL	d (m)	1.14 ± 0.70	1.34 ± 0.01	0.001	0.404
		t (s)	0.98 ± 0.16	0.75 ± 0.17	−1.225	1.393
	TDPL	d (m)	3.04 ± 1.70	4.23 ± 0.11	2.589 *	0.987
		t (s)	2.47 ± 0.28	3.57 ± 0.28	−2.775 *	3.928
720°	DPB	d (m)	2.87 ± 0.78	3.36 ± 0.44	−2.917 *	0.773
		t (s)	1.95 ± 0.21	2.27 ± 0.42	−2.768 *	0.963
	DSL	d (m)	1.28 ± 0.96	1.44 ± 0.41	−1.091	0.216
		t (s)	1.10 ± 0.00	0.90 ± 0.00	−1.633	0.216
	TDPL	d (m)	3.15 ± 1.75	4.80 ± 0.85	−2.675 *	1.199
		t (s)	3.05 ± 0.21	4.17 ± 0.42	−2.901 *	3.373

All data represent mean ± standard deviation (SD). LESS IG, a group with less than 20 months of injury experience; MORE IG, a group with more than 20 months of injury experience; ES, effect size; d, distance; t, time. DPB, Distance from prior jump to before jump; DSL, Distance from spinning to landing; TDPL, Total distance from take-off to landing. * *p* < 0.05.

**Table 4 medicina-57-01166-t004:** Comparison of contraction time from TMG of lower legs.

Measured Muscles	TMG Variables	Groups			
		LESS IG	MORE IG	Z	ES
Biceps Femoris	Tc (ms) of Lt	21.42 ± 0.13	23.63 ± 4.57	−0.684	0.683
	Dm (mm) of Lt	3.27 ± 2.69	2.31 ± 0.67	0.493	0.489
	Tc (ms) of Rt	22.68 ± 3.90	19.61 ± 6.99	0.543	0.542
	Dm (mm) of Rt	1.09 ± 0.42	4.85 ± 0.40	−5.425 **	9.168
Gastrocnemius Lateralis	Tc (ms) of Lt	26.82 ± 11.19	25.80 ± 3.06	0.125	0.124
	Dm (mm) of Lt	2.11 ± 0.74	4.25 ± 0.62	−4.504 **	3.134
	Tc (ms) of Rt	32.69 ± 3.76	16.23 ± 24.13	2.957 *	0.953
	Dm (mm) of Rt	1.87 ± 1.73	5.79 ± 1.56	−2.784 *	2.379
Gastrocnemius Medialis	Tc (ms) of Lt	24.04 ± 11.82	20.55 ± 0.03	0.417	0.417
	Dm (mm) of Lt	1.65 ± 0.62	1.92 ± 0.08	−0.608	0.610
	Tc (ms) of Rt	37.17 ± 26.65	15.82 ± 2.61	3.428 *	1.127
	Dm (mm) of Rt	0.72 ± 0.61	0.96 ± 0.76	−1.115	0.348
Rectus Femoris	Tc (ms) of Lt	21.99 ± 0.27	13.74 ± 0.25	4.418 **	31.707
	Dm (mm) of Lt	4.37 ± 1.15	8.80 ± 1.78	−2.376 *	2.956
	Tc (ms) of Rt	18.33 ± 5.43	18.75 ± 2.01	−0.103	0.102
	Dm (mm) of Rt	4.34 ± 1.48	4.74 ± 0.06	−0.376	0.381
Tibialis Anterior	Tc (ms) of Lt	29.01 ± 2.84	17.28 ± 6.06	2.746 *	2.478
	Dm (mm) of Lt	0.83 ± 0.42	1.03 ± 0.37	−0.504	0.505
	Tc (ms) of Rt	49.17 ± 0.49	22.86 ± 42.41	2.990 *	0.877
	Dm (mm) of Rt	2.07 ± 0.76	2.11 ± 1.73	−0.030	0.029
Vastus Lateralis	Tc (ms) of Lt	22.73 ± 5.56	20.94 ± 1.07	0.301	0.447
	Dm (mm) of Lt	3.20 ± 2.88	4.92 ± 1.66	−0.732	0.731
	Tc (ms) of Rt	19.00 ± 10.92	18.78 ± 2.84	0.348	0.027
	Dm (mm) of Rt	3.43 ± 4.54	3.41 ± 0.42	0.006	0.006
Vastus Medialis	Tc (ms) of Lt	29.94 ± 1.75	17.56 ± 3.67	2.916 *	4.306
	Dm (mm) of Lt	2.26 ± 2.75	5.56 ± 0.25	−1.383	1.690
	Tc (ms) of Rt	22.74 ± 6.81	25.14 ± 3.85	−0.434	0.433
	Dm (mm) of Rt	4.46 ± 0.56	4.70 ± 0.86	−0.332	0.330

All data represent mean ± standard deviation (SD). LESS IG, a group with less than 20 months of injury experience; MORE IG, a group with more than 20 months of injury experience; ES, effect size; TMG, tensiomyography; Tc, contraction time; Dm, maximal displacement. *, *p* < 0.05; **, *p* < 0.01.

**Table 5 medicina-57-01166-t005:** Comparison results of isokinetic knee extension/flexion at 60°/s, ankle inversion/eversion at 60°/s, and ankle plantarflexion/dorsiflexion at 30°/s.

Measured Muscles	Pt/Wr of Sides	Groups			
		LESS IG	MORE IG	Z	ES
Knee Extensor	Pt (Nm) of Rt	194.00 ± 69.30	145.50 ± 13.44	0.972	0.971
	Pt (Nm) of Lt	220.00 ± 29.70	156.50 ± 21.92	2.433 *	2.432
	Wr (Nm) of Rt	174.50 ± 40.31	145.50 ± 10.61	0.984	0.983
	Wr (Nm) of Lt	208.00 ± 14.14	151.00 ± 29.70	2.451 *	2.450
Knee Flexor	Pt (Nm) of Rt	152.00 ± 48.08	105.50 ± 7.78	1.350	1.350
	Pt (Nm) of Lt	156.50 ± 4.95	96.00 ± 24.04	3.486 **	3.485
	Wr (Nm) of Rt	158.00 ± 8.49	125.00 ± 24.04	1.831	1.830
	Wr (Nm) of Lt	167.00 ± 22.63	102.00 ± 29.70	2.462 *	2.461
Ankle Invertor	Pt (Nm) of Rt	18.00 ± 0.01	18.50 ± 0.71	−1.000	0.995
	Pt (Nm) of Lt	23.00 ± 5.66	16.00 ± 2.83	1.565	1.564
	Wr (Nm) of Rt	10.00 ± 1.41	11.00 ± 0.01	−1.000	1.002
	Wr (Nm) of Lt	15.50 ± 6.36	11.00 ± 0.01	1.000	1.000
Ankle Evertor	Pt (Nm) of Rt	17.00 ± 2.83	16.50 ± 3.54	0.156	0.156
	Pt (Nm) of Lt	19.00 ± 0.01	17.50 ± 2.12	1.000	1.000
	Wr (Nm) of Rt	11.00 ± 0.01	10.50 ± 2.12	0.333	0.333
	Wr (Nm) of Lt	14.50 ± 4.95	13.00 ± 1.41	0.412	0.412
Ankle Plantarflexor	Pt (Nm) of Rt	123.00 ± 14.14	104.00 ± 8.49	2.629 *	1.629
	Pt (Nm) of Lt	101.50 ± 7.78	105.00 ± 8.49	−0.430	0.429
	Wr (Nm) of Rt	71.00 ± 2.83	43.50 ± 13.44	2.833 *	2.831
	Wr (Nm) of Lt	50.50 ± 4.95	47.50 ± 2.12	0.788	0.787
Ankle Dorsiflexor	Pt (Nm) of Rt	37.00 ± 5.66	39.00 ± 5.66	−0.354	0.353
	Pt (Nm) of Lt	39.50 ± 3.54	33.50 ± 4.95	1.395	1.394
	Wr (Nm) of Rt	24.50 ± 7.78	17.00 ± 1.41	1.342	1.341
	Wr (Nm) of Lt	24.00 ± 5.66	15.00 ± 4.24	1.800	1.799

All data represent mean ± standard deviation. LESS IG, a group with less than 20 months of injury experience; MORE IG, a group with more than 20 months of injury experience; ES, effect size; Pt, peak torque; Wr, work per repetition; Rt, right; Lt, left. *, *p* < 0.05; **, *p* < 0.01.

**Table 6 medicina-57-01166-t006:** Comparison results of isokinetic knee extension/flexion at 180°/s, ankle inversion/eversion at 90°/s, and ankle plantarflexion/dorsiflexion at 120°/s.

Measured Muscles	Pt/Fi/Tw of Sides	Groups			
		LESS IG	MORE IG	Z	ES
Knee Extensor	Pt (Nm) of Rt	131.00 ± 35.36	110.50 ± 6.36	0.807	0.806
	Pt (Nm) of Lt	147.50 ± 16.26	95.50 ± 37.48	2.800 *	1.799
	Fi of Rt	10.00 ± 9.90	8.50 ± 0.71	0.214	0.213
	Fi of Lt	20.50 ± 7.78	−2.50 ± 20.51	3.483 **	1.482
	Tw (Nm) of Rt	1661.00 ± 65.05	1652.50 ± 133.64	0.081	0.080
	Tw (Nm) of Lt	1824.50 ± 65.76	1350.50 ± 525.38	2.566 *	1.266
Knee Flexor	Pt (Nm) of Rt	110.50 ± 4.95	83.50 ± 6.36	2.736 *	4.737
	Pt (Nm) of Lt	122.50 ± 10.61	68.50 ± 27.58	3.585 **	2.584
	Fi of Rt	9.50 ± 2.12	9.00 ± 19.80	0.036	0.035
	Fi of Lt	14.00 ± 1.41	−8.00 ± 46.67	1.666	0.666
	Tw (Nm) of Rt	1504.00 ± 333.75	1266.50 ± 125.16	2.942 *	0.942
	Tw (Nm) of Lt	1614.50 ± 365.57	1028.00 ± 258.80	2.852 *	1.851
Ankle Invertor	Pt (Nm) of Rt	16.00 ± 0.01	14.50 ± 0.71	1.256	2.987
	Pt (Nm) of Lt	16.00 ± 2.83	15.00 ± 4.24	0.277	0.274
	Fi of Rt	21.00 ± 18.38	4.50 ± 13.44	3.025 **	1.024
	Fi of Lt	5.00 ± 1.41	21.50 ± 23.33	−3.998 **	0.998
	Tw (Nm) of Rt	124.00 ± 19.80	135.00 ± 19.80	−0.556	0.555
	Tw (Nm) of Lt	168.50 ± 75.66	125.50 ± 6.36	1.801	0.800
Ankle Evertor	Pt (Nm) of Rt	16.00 ± 2.83	13.00 ± 2.83	1.061	1.060
	Pt (Nm) of Lt	17.00 ± 1.41	16.50 ± 3.54	0.186	0.185
	Fi of Rt	28.00 ± 4.24	19.50 ± 4.95	1.844	1.844
	Fi of Lt	24.00 ± 8.49	38.00 ± 2.83	−1.972	2.212
	Tw (Nm) of Rt	122.00 ± 19.80	106.50 ± 12.02	0.946	0.944
	Tw (Nm) of Lt	169.50 ± 61.52	126.00 ± 1.41	2.030	0.999
Ankle Plantarflexor	Pt (Nm) of Rt	72.00 ± 11.31	54.50 ± 7.78	1.803	0.021
	Pt (Nm) of Lt	62.00 ± 8.49	52.50 ± 4.95	1.368	1.367
	Fi of Rt	17.00 ± 18.38	21.00 ± 1.41	−0.307	0.306
	Fi of Lt	9.50 ± 20.51	21.50 ± 7.78	−0.774	0.773
	Tw (Nm) of Rt	594.00 ± 48.08	322.50 ± 137.89	4.629 **	2.629
	Tw (Nm) of Lt	520.50 ± 159.10	372.50 ± 126.57	2.730 *	1.029
Ankle Dorsiflexor	Pt (Nm) of Rt	22.50 ± 0.71	25.00 ± 4.24	−0.822	1.151
	Pt (Nm) of Lt	26.00 ± 0.01	23.50 ± 0.71	5.000	2.987
	Fi of Rt	18.00 ± 5.66	22.00 ± 29.70	−0.187	0.233
	Fi of Lt	14.00 ± 11.31	20.00 ± 7.07	−0.636	0.742
	Tw (Nm) of Rt	186.50 ± 17.68	153.50 ± 36.06	1.866	1.174
	Tw (Nm) of Lt	198.50 ± 58.69	152.50 ± 33.23	1.965	0.964

All data represent mean ± standard deviation. LESS IG, a group with less than 20 months of injury experience; MORE IG, a group with more than 20 months of injury experience; ES, effect size; Pt, peak torque; Fi, fatigue index; Tw, total work; Rt, right; Lt, left. *, *p* < 0.05; **, *p* < 0.01.

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
