# Peer review of "Effects of Knee Injury Length on Jump Inside Kick Performances of Wushu Player"

_medicina, 2021, doi:10.3390/medicina57111166_

Round 1

Reviewer 1 Report

It is interesting to see that the two groups of Wushu athletes with different injury experience showed differences only in the performance with high difficulty. However, there are a few major points to consider for this manuscript.

  • Title, definitions of key words

What is ‘frequency-Effects’ of injury? Similarly, what is the definition of 20 month of ‘injury experience’ (manuscript)? How about ‘Injury period’ (table)? Need clarification.

  • Focus of the manuscript

Judging from the title, the focus should be on the differences in performance and potential explanations. First of all, does ‘...a shorter jump distance but a faster rotational force…’ describes the desired movement pattern in this task? Incidentally, ‘a faster rotational force’ does not make sense. The authors have not measured force and there is no ‘faster’ force.

As a result of the performance analysis, the authors indicated that a conversion from forward speed to vertical speed is a key that accounts for the differences in performance between two groups. Therefore, following discussion should be around this point. Currently, the focus of the discussion section is dilated and hard to follow. There is almost one page long paragraph in discussion session.

  • Presentation of the results

The large tables filled with numbers are simply hurled at the readers. Most of the contents are not even mentioned in the result section, let alone in the discussed session. Some pieces of information in the tables are duplicated in the manuscript of the result section as well as the discussion section. It really hampered the readability of this manuscript. 

Author Response

Answers to reviewer’s comments 

Thank you for your kind advice and comments for publication in Medicina. We revised our manuscript as per your comments. We represented the specific modifications in response to the comments by blue-letters in our manuscript. We sincerely appreciate your comments because your comments make our manuscript better.

Comments and Suggestions for Authors

It is interesting to see that the two groups of Wushu athletes with different injury experience showed differences only in the performance with high difficulty. However, there are a few major points to consider for this manuscript.

  • Title, definitions of key words
  1. What is ‘frequency-Effects’ of injury?
  2. Similarly, what is the definition of 20 month of ‘injury experience’ (manuscript)?
  3. How about ‘Injury period’ (table)? Need clarification.

#Response 1: Thank you for what the reviewer has pointed out the comment. In the title, ...'frequency-Effects'... meant the frequency of injuries, but I think it was awkward like your opinion, so I changed it to ...Effects of knee injury frequency. Ultimately, the revised title is as follows, and the text is also modified in the same way.

 “Effects of Knee Injury Frequency on Jump Inside Kick Performances of Wushu Player”

#Response 2: Thank you for what the reviewer has pointed out the comment. The reason for classifying the group as ‘20 months’ was the application of the Cochran–Mantel–Haenszel equation. In other words, the injury risk was expressed as the number of knee injured as a percentage of the total number. The Cochran–Mantel–Haenszel risk ratio can be computed as follows:

It is a value calculated by applying this formula considering the ratio of injuries to total athletic experience (Mantel and Haenszel, 1959). Moreover, the subjects of this study were evaluated by a specialist, and athletes with an injury frequency of 20 months or more were classified as having a high risk of repeated injury, whereas those of less than 20 months were classified as having a moderate injury frequency. To reduce confusion, the following sentences are inserted into the text by organizing the above contents.

“The reason for classifying the group as ‘20 months’ was the application of the Cochran–Mantel–Haenszel equation, which the injury risk was expressed as the number of knee injured as a percentage of the total number (Mantel and Haenszel, 1959). The subjects of this study were evaluated by a specialist, and athletes with an injury frequency of 20 months or more were classified as having a high risk of repeated injury, whereas those of less than 20 months were classified as having a moderate injury frequency.

[32] Mantel, N.; Haenszel, W. Statistical aspects of the analysis of data from retrospective studies of disease. J Natl Cancer Inst 1959, 22, 719-748.”

#Response 3: Thank you for what the reviewer has pointed out the comment. ‘Injury period’ refers to the period of injury during the athletic career of Wushu athletes assigned to each group.

  • Focus of the manuscript
  1. Judging from the title, the focus should be on the differences in performance and potential explanations. First of all, does ‘...a shorter jump distance but a faster rotational force…’ describes the desired movement pattern in this task? Incidentally, ‘a faster rotational force’ does not make sense. The authors have not measured force and there is no ‘faster’ force.
  2. As a result of the performance analysis, the authors indicated that a conversion from forward speed to vertical speed is a key that accounts for the differences in performance between two groups. Therefore, following discussion should be around this point. Currently, the focus of the discussion section is dilated and hard to follow. There is almost one page long paragraph in discussion session.

#Response 4: Thank you for what the reviewer has pointed out the comment. As shown in Fig. 1 and Fig. 2, five events and four phases were used for the kinematic analysis of three motions of the jump inside kick. Five events and four phases include spinning time. Therefore, we use the phrase ‘a faster rotational force’ in our paper. If you are confused, I will replace ‘a faster rotational force’ with ‘a faster rotational time’. In the text of our study, 'force' was changed to 'time'.

“…..significantly higher in LESS IG than in MORE IG, which is interpreted to suggest the importance of the right plantarflexor that triggers the jumping force time to the left.’

#Response 5: Thank you for what the reviewer has pointed out the comment. We have revised the results based on your comments, and most of the discussion has been revised as follows.

“In the discussion part….. such a fast rotation time must be supported by the myofunction capable of exerting force..’

“….more technical training is needed at over 360° spinning skill such as shorter spinning time..’

  • Presentation of the results
  1. The large tables filled with numbers are simply hurled at the readers. Most of the contents are not even mentioned in the result section, let alone in the discussed session. Some pieces of information in the tables are duplicated in the manuscript of the result section as well as the discussion section. It really hampered the readability of this manuscript. 

#Response 6: Thank you for what the reviewer has pointed out above comments. After listening to your comments and looking at our results, the interpretation of the results and the figures in the table often overlap. So, according to your opinion, the figures presented in the results and discussion have been deleted or only statistical meanings have been presented, and the sentences are as follows.

“Table 2 shows that the total event time from E1 to E5 at 360° in the LESS IG appeared 0.04 sec longer than that of the MORE IG. On the other hand, at 540°, the total event time from E1 to E5 in the LESS IG was 0.27 sec shorter than that of the MORE IG, indicating that ability of LESS IG to rotate in the air was faster than that of MORE IG.....

Table 2 also shows that the total phase times at 360° and at 540° did not show a significant difference between the two groups, but at 720° rotation, it was confirmed that LESS IG was significantly faster than MORE IG.

Table 3 represents the distance and time from take-off to landing between LESS IG and MORE IG. For the total distance covered in the 360° jump inside kick, LESS IG was shorter than MORE IG. This trend was similar in 540° and 720° rotation, and it can be observed that there is a statistically significant difference between groups.”

Thank you for your comments, we represented the modifications in response to your comments.

October 21, 2021

Reviewer 2 Report

First of all, the reviewer would like to thank the authors for their work and efforts in trying to improve sports science knowledge. The article is an interesting approach to the Frequency-Effects of Knee Injury on Jump Inside Kick Performances of Wushu Player. The study is well designed and well-written, with a great introduction proposing the usefulness of the topic and a clear outline of the research question.

I suggest that the author modify / include some suggestions in order to improve the manuscript.

Author Response

Answers to reviewer’s comments 

Thank you for your kind advice and comments for publication in Medicina. We revised our manuscript as per your comments. We represented the specific modifications in response to the comments by blue-letters in our manuscript. We sincerely appreciate your comments because your comments make our manuscript better.

Comments and Suggestions for Authors

First of all, the reviewer would like to thank the authors for their work and efforts in trying to improve sports science knowledge. The article is an interesting approach to the Frequency-Effects of Knee Injury on Jump Inside Kick Performances of Wushu Player. The study is well designed and well-written, with a great introduction proposing the usefulness of the topic and a clear outline of the research question.

I suggest that the author modify / include some suggestions in order to improve the manuscript.

#Response: Thank you for what the reviewer has pointed out above comments. I looked up the PDF file you requested for correction. It is judged that the effect size is inserted in all tables. So, we turned the statistics again and inserted the effect size into all tables as follows (ref. attached file).

Reviewer 3 Report

Main comments and considerations 

Very nteresting manuscript. Especially that in the available literature there are not too many items with such a (multi-threaded) analysisIn the available literature, there are few analyzes given on comparable groups of martial arts sports. However, it requires some corrections and more details.

First, the text does not explain how the knee injury frequency-effects affects on Jump Inside Kick Performances of Wushu Player and this is the title of this manuscript. Therefore, this relationship should be clarified more or the title should be changed.

It is necessary to describe in more detail what the authors understand by the terms LESS IG and MORE IG, because in the present form of the description it is not unequivocal.

Less or More than 20 months of injury experience means that competitors have had knee injuries for less or more than 20 months for a total of 20 months throughout their career or within 20 months from the injury to the end of treatment and rehabilitation - on a time continuum - before returning to sport and training?  This is not clearly explained ! Please correct it in the description of participants (2.1)  

Where the authors show the frequency of injuries and how they used it to divide research subgroups? Such a description of the division of research groups rather leads to the topic of the impact of the time from injury to return to sport (including time with a knee injury, treatment and rehabilitation) on the performance of Jump Inside Kick and its functional characteristics (physiological and biomechanical). It is the effect of the length of the injury rather than the frequency of the injuries!  

The research itself was carried out with due diligence and in accordance with high methodological standards.  However, since the analysis of body composition using the method of electrical bioimpedance was performed, why not more parameters from this analysis were shown? Such an analysis would be interesting, with particular emphasis on the differences between the limbs which were injured and which were not. The more so because, as it was written in the methods, segmental analysis was performed (only body weight and height are shown).

The results would be more valuable if all the quantities characterizing the kinematics of movement and kinetics were shown with the division into the limb with and without knee injury (and not only for Rt and Lt leg). Perhaps the authors would like to supplement this work with such a division and analysis?

There were also no important information on whether the players were right or left-handed and whether they performed Wushu techniques for analysis with rotation in the same direction (e.g. all to the right or all to the left, or maybe both directions)

Were Inbody measurements taken at the same time of day for everyone, preferably in the morning and fasting? Electrical bioimpedance measurements are very sensitive, and competitors should avoid physical activity for a minimum of 15-20 minutes prior to the measurement. Even climbing the stairs to the laboratory can disrupt them. Therefore, before the measurement, they should remain at rest, in a comfortable position, without any physical activity.    

With such small research groups, I recommend using the Will Hopkins' MBI statistics rather than using the significance value of p (P Values vs Magnitude-based Inference) - http://sportsci.org/ )   

Line 105 and further - expanding ACL and PCL at least once   

Finally, the limitations of this work need to be described in more detail 

First of all, the lack of a control group (composed of players without knee injuries) significantly worsens the analysis of the obtained results, because there is no "healthy" point of reference. 

Secondly (as the authors emphasized) a small research group requires more caution in making conclusions. I am aware of the fact that it is extremely difficult to find a much larger number of players with a such high sports level that allows to make jumps with a kick and rotation at the level of 540 or 720 degrees. Especially with a large number of additional study exclusion criteria (as described by the authors).But it needs to be emphasized and explained. Therefore, I would suggest adding a short chapter after the chapter conclusions "The limitations of the study”. 

I think that despite my comments regarding errors or limitations, the paper is interesting and worth corrections and improvements.

Author Response

Answers to reviewer’s comments 

Thank you for your kind advice and comments for publication in Medicina. We revised our manuscript as per your comments. We represented the specific modifications in response to the comments by blue-letters in our manuscript. We sincerely appreciate your comments because your comments make our manuscript better.

Comments and Suggestions for Authors

Main comments and considerations 

Very nteresting manuscript. Especially that in the available literature there are not too many items with such a (multi-threaded) analysis In the available literature, there are few analyzes given on comparable groups of martial arts sports. However, it requires some corrections and more details.

  1. First, the text does not explain how the knee injury frequency-effects affects on Jump Inside Kick Performances of Wushu Player and this is the title of this manuscript. Therefore, this relationship should be clarified more or the title should be changed.

#Response 1: Thanks for your meticulous comments and suggestions. In the title, ...'frequency-Effects'... meant the frequency of injuries, but I think it was awkward like your opinion, so I changed it to ...Effects of knee injury frequency. Ultimately, the revised title is as follows, and the text is also modified in the same way.

 “Effects of Knee Injury Frequency on Jump Inside Kick Performances of Wushu Player”

  1. It is necessary to describe in more detail what the authors understand by the terms LESS IG and MORE IG, because in the present form of the description it is not unequivocal.

#Response 2: Thank you for what the reviewer has pointed out the comment. The reason for classifying the group as ‘20 months’ was the application of the Cochran–Mantel–Haenszel equation (Mantel and Haenszel, 1959). In other words, the injury risk was expressed as the number of knee injured as a percentage of the total number. The Cochran–Mantel–Haenszel risk ratio can be computed as follows:

In addition, the subjects of this study were evaluated by a specialist, and athletes with an injury frequency of 20 months or more were classified as having a high risk of repeated injury, whereas those of less than 20 months were classified as having a moderate injury frequency. To reduce confusion, the following sentences are inserted into the text by organizing the above contents.

“The reason for classifying the group as ‘20 months’ was the application of the Cochran–Mantel–Haenszel equation, which the injury risk was expressed as the number of knee injured as a percentage of the total number (Mantel and Haenszel, 1959). The subjects of this study were evaluated by a specialist, and athletes with an injury frequency of 20 months or more were classified as having a high risk of repeated injury, whereas those of less than 20 months were classified as having a moderate injury frequency.

[32] Mantel, N.; Haenszel, W. Statistical aspects of the analysis of data from retrospective studies of disease. J Natl Cancer Inst 1959, 22, 719-748.”

  1. Less or More than 20 months of injury experience means that competitors have had knee injuries for less or more than 20 months for a total of 20 months throughout their career or within 20 months from the injury to the end of treatment and rehabilitation - on a time continuum - before returning to sport and training?  This is not clearly explained ! Please correct it in the description of participants (2.1)  

#Response 3: Thanks for your meticulous comments and suggestions. Question #3 was answered in question #2 as follows: the reason for classifying the group as ‘20 months’ was the application of the Cochran–Mantel–Haenszel equation (Mantel and Haenszel, 1959). The injury risk was expressed as the number of knee injured as a percentage of the total number.

  1. Where the authors show the frequency of injuries and how they used it to divide research subgroups? Such a description of the division of research groups rather leads to the topic of the impact of the time from injury to return to sport (including time with a knee injury, treatment and rehabilitation) on the performance of Jump Inside Kick and its functional characteristics (physiological and biomechanical). It is the effect of the length of the injury rather than the frequency of the injuries!  

#Response 4: Thank you for what the reviewer has pointed out the comment. Based on the reviewers' comments, it seems that the intention of our paper is the length of the injury rather than the frequency of the injury. So let's change the title to the following:

“Effects of Knee Injury Length on Jump Inside Kick Performances of Wushu Player”

In our country, it is generally decided solely by the orthopedic surgeon for an athlete to return to the field after receiving treatment after an injury. We presented the text in consideration of only what kind of treatment Wushu received after being hospitalized or when he returned. I hope you understand widely.

  1. The research itself was carried out with due diligence and in accordance with high methodological standards.  However, since the analysis of body composition using the method of electrical bioimpedance was performed, why not more parameters from this analysis were shown? Such an analysis would be interesting, with particular emphasis on the differences between the limbs which were injured and which were not. The more so because, as it was written in the methods, segmental analysis was performed (only body weight and height are shown).

#Response 5: Thanks for your meticulous comments and suggestions. Of course, we also have a lot of data because we measured body composition with electrical bioimpedance as you suggested. However, what we were trying to focus on was to observe how badly the damage was in the performance area of ​​Wushu players. Therefore, please understand that detailed data derived from body composition have not been utilized.

  1. The results would be more valuable if all the quantities characterizing the kinematics of movement and kinetics were shown with the division into the limb with and without knee injury (and not only for Rt and Lt leg). Perhaps the authors would like to supplement this work with such a division and analysis?

#Response 6: Thanks for your meticulous comments and suggestions. The points pointed out by the reviewer have already been described in the research method as follows.

“In LESS IG, three patients were diagnosed with right ACL rupture, but recovered after 5, 7, and 8 months of rehabilitation treatment, respectively. On the other hand, one player was diagnosed with a right hamstrings’ sprain and received 5 months of treatment, while the other two player s recovered after 18 months and 20 months of postoperative rehabilitation treatment for partial rupture of the right knee joint cartilage, respectively. In MORE IG, two player s were diagnosed with a simple right ACL rupture and two players were diagnosed with ACL+PCL complex rupture of the right knee joint, and recovered after 7, 14, 16, and 18 months of rehabilitation treatment, respectively. Meanwhile, one player was diagnosed with right hamstring’s rupture and recovered after 12 months of treatment. However, 7 days later, during training, he ruptured again in the same area and had to undergo treatment for another 12 months. The other player was diagnosed with an ACL+PCL complex rupture of the right knee and recovered after 18 months including surgery and rehabilitation. However, this athlete recovered after 18 months of rehabilitation after surgery due to a partial rupture of the right knee joint cartilage during training.”

  1. There were also no important information on whether the players were right or left-handed and whether they performed Wushu techniques for analysis with rotation in the same direction (e.g. all to the right or all to the left, or maybe both directions)

#Response 7: Thanks for your meticulous comments and suggestions. They were all right-handed and right-footed. And Wushu's jump inside kick rotates from left to right. Of course, there are a few players who rotate in the opposite direction, but in Korea, wushu players all take the jump inside kick in a counterclockwise direction. Based on your opinion, 'they were all right-footed and right-footed' was inserted into the sentence.

  1. Were Inbody measurements taken at the same time of day for everyone, preferably in the morning and fasting? Electrical bioimpedance measurements are very sensitive, and competitors should avoid physical activity for a minimum of 15-20 minutes prior to the measurement. Even climbing the stairs to the laboratory can disrupt them. Therefore, before the measurement, they should remain at rest, in a comfortable position, without any physical activity.

#Response 8: Thanks for your meticulous comments and suggestions. The reviewer is right. So we kept everything and measured it in the morning before the activity. The following sentence has been inserted to avoid confusion.

'Bioelectrical resistance analysis method can track fat mass by conducting high frequency (500~800KHz) harmless to the human body and using the difference in electrical resistance between adipose tissue and non-adipose tissue. In order to minimize the error, in this study, food intake was abstained 4 hours before the test, and alcohol was prohibited 48 hours before the test. In addition, exercise was not allowed 12 hours before the test. On the other hand, it was necessary to urinate 30 minutes before the test.'

  1. With such small research groups, I recommend using the Will Hopkins' MBI statistics rather than using the significance value of p (P Values vs Magnitude-based Inference) - http://sportsci.org/ )   

#Response 9: Thanks for your meticulous comments and suggestions. However, I do not have the statistical program you suggested and I have not learned it yet. If there is a chance next time, I will definitely use this program for a small number of subjects. Thanks again for the good recommendations and advice.

  1. Line 105 and further - expanding ACL and PCL at least once   

#Response 10: I have no idea what this point means. Line 105 and below list the injuries suffered by the subjects in this study.

  1. Finally, the limitations of this work need to be described in more detail. First of all, the lack of a control group (composed of players without knee injuries) significantly worsens the analysis of the obtained results, because there is no "healthy" point of reference. Secondly (as the authors emphasized) a small research group requires more caution in making conclusions. I am aware of the fact that it is extremely difficult to find a much larger number of players with a such high sports level that allows to make jumps with a kick and rotation at the level of 540 or 720 degrees. Especially with a large number of additional study exclusion criteria (as described by the authors).But it needs to be emphasized and explained. Therefore, I would suggest adding a short chapter after the chapter conclusions "The limitations of the study”. 

#Response 11: Thanks for your meticulous comments and suggestions. After collecting all your opinions, the limitations of this study are described as follows.

“However, this study had a small sample size due to the limited number of Wushu players in Korea. In addition, healthy Wushu players were not included in the study. Therefore, there are limitations in being able to generalize these results to other populations and types of competitions.”

Thank you for your comments, we represented the modifications in response to your comments.

October 21, 2021

Round 2

Reviewer 1 Report

Thank you for a quick revision. Some of my comments are addressed to my satisfaction. Let me leave final comments. What to do with them is at the authors' discretion. 

1) Although I still don't understand why the authors needed to present ALL possible information to support the conclusion, I suppose this is the style the authors prefer. 

2) Although I don't observe any conflicts between the results and following discussion, some of them are not tightly coupled with this specific task. For example, muscle strength, endurance and power indicate different aspects of muscle capabilities.  Which one is more important than the others in this specific task? And how are they related to static/dynamic muscle contractility? I wished I could seen the straight answers to these questions. 

Author Response

Answers to reviewer’s comments (2nd Revision) 

Thank you for your kind advice and comments for publication in Medicina. We re-revised our manuscript as per your comments. We represented the specific modifications in response to the comments by blue-letters in our manuscript. We sincerely appreciate your comments because your comments make our manuscript better.

Comments and Suggestions for Authors

1) Although I still don't understand why the authors needed to present ALL possible information to support the conclusion, I suppose this is the style the authors prefer. 

#Response 1: Thank you for what the reviewer has pointed out the comment. Ultimately, this study found that when Wushu players rotate 360, 540, and 720 degrees in the air, the rotation speed varies according to the three target rotation angles, and there is a significant difference according to the length of the injury. However, when the length of the injuries of the athletes is long, the muscle function of the lower extremities to cause rotation in the air is low, and moreover, when the number of rotations is high, it is more evident. These results suggest that even if an athlete is injured, treatment should be carried out quickly so that the frequency or length of the injury does not occur as often.

2) Although I don't observe any conflicts between the results and following discussion, some of them are not tightly coupled with this specific task. For example, muscle strength, endurance and power indicate different aspects of muscle capabilities.  Which one is more important than the others in this specific task? And how are they related to static/dynamic muscle contractility? I wished I could seen the straight answers to these questions. 

#Response 2: Thank you for what the reviewer has pointed out the comment. In the jump inside kick observed in this study, power is more important than muscle strength or endurance because instantaneous force must be generated before jumping into the air. Sequentially, muscle strength is important to generate reflexes, and muscular endurance is required to continuously exert repetitive force. In order to investigate these factors as much as possible, this study observed peak torque, work per repetition, fatigue index and total work in dynamic muscle function. Of course, static muscle strength was measured and observed to examine the muscle function state first before performing dynamic movements. Although there is a high correlation between static muscle contraction and dynamic muscle contraction, I thought that the static muscle contraction test should be preceded prior to the dynamic test or rotational motion because all the players I wanted to analyze had experienced injuries. I hope this answers your question.

Thank you for your comments again, I represented the modifications in response to your comments.

October 25, 2021
